# Differences in Metabolite Composition of *Aloe barbadensis* Mill. Extracts Lead to Differential Effects on Human Blood T Cell Activity In Vitro

**DOI:** 10.3390/molecules27196643

**Published:** 2022-10-06

**Authors:** Bani Ahluwalia, Maria K. Magnusson, Fredrik Larsson, Otto Savolainen, Alastair B. Ross, Lena Öhman

**Affiliations:** 1Department of Microbiology and Immunology, Institute of Biomedicine, University of Gothenburg, 405 30 Gothenburg, Sweden; 2Research and Development, Calmino Group AB, 413 46 Gothenburg, Sweden; 3Chalmers Mass Spectrometry Infrastructure, Department of Biology and Biological Engineering, Chalmers University of Technology, 412 96 Gothenburg, Sweden; 4Faculty of Health Sciences, Institute of Public Health and Clinical Nutrition, University of Eastern Finland, 70211 Kuopio, Finland; 5Proteins and Metabolites Team, AgResearch, Lincoln 7674, New Zealand

**Keywords:** Aloe, metabolomics, T cells activity, bioactive composition

## Abstract

*Aloe barbadensis* Mill. (Aloe) is used for diverse therapeutic properties including immunomodulation. However, owing to the compositionally complex nature of Aloe, bioactive component(s) responsible for its beneficial properties, though thought to be attributed to polysaccharides (acemannan), remain unknown. We therefore aimed to determine the metabolite composition of various commercial Aloe extracts and assess their effects on human blood T cell activity in vitro. Peripheral blood mononuclear cells (PBMC) from healthy donors were stimulated polyclonally in presence or absence of various Aloe extracts. T cell phenotype and proliferation were investigated by flow cytometry. Aloe extracts were analyzed using targeted ^1^H-NMR spectroscopy for standard phytochemical quality characterization and untargeted gas chromatography mass spectrometry (GC-MS) for metabolite profiling. Aloe extracts differing in their standard phytochemical composition had varying effects on T cell activation, proliferation, apoptosis, and cell-death in vitro, although this was not related to the acemannan content. Furthermore, each Aloe extract had its own distinct metabolite profile, where extracts rich in diverse sugar and sugar-derivatives were associated with reduced T cell activity. Our results demonstrate that all commercial Aloe extracts are unique with distinct metabolite profiles, which lead to differential effects on T cell activity in vitro, independent of the acemannan content.

## 1. Introduction

*Aloe barbadensis* Mill. or Aloe (family Xanthorrhoeaceae, subfamily Asphodeloideae), a medicinal plant with a long history of use [1], continues to generate interest in the food and pharmaceutical industry for the development of functional foods and therapeutic products [2]. Aloe gel, the colorless mucilaginous gel from inner parenchymatous Aloe leaf, is a rich source of bioactive compounds and is thought to be the main source of attributed therapeutic benefits [3,4,5,6,7]. Many of the properties of the Aloe gel have been ascribed to its complex polysaccharide composition, in particular acetylated mannose, also called acemannan [8,9,10]. Besides being the main as well as the most studied polysaccharide, acemannan is used as one of the quality markers for defining Aloe products [11]. Additionally, studies have correlated beneficial properties of Aloe to high-molecular weight polysaccharide fractions, highlighting the importance of molecular weight of Aloe to its therapeutic effects [12]. Nevertheless, despite extensive use and several scientific studies over the last decades, there is no clear correlation between the beneficial properties of Aloe gel and its composition, and the debate regarding the component or group of components accountable for the various therapeutic properties remains unresolved.

The effect of Aloe on immune cells has been studied in various models, showing encouraging results for its beneficial immunomodulatory properties [6,13,14,15,16,17]. However, the studies supporting these positive effects are inconsistent in the form or part of Aloe used, making it difficult to correlate the effect with a specific component or composition of Aloe. Additionally, results from an in vitro study performed by our group [13], evaluating the effect of two Aloe extracts on T cell activity, demonstrated that extracts with distinct chemical compositions differed in their potency for suppressive effect. The quantitative and qualitative variation in biologically active or “bioactive” composition of Aloe and consequently the efficacy of Aloe products (liquid and dehydrated extracts), are known to be influenced by several factors including geographical location, seasonal changes and most importantly processing and extraction methods [3,11,18]. Hence, further studies with emphasis on the complex nature of Aloe composition before and after processing to improve the understanding of its bioactive composition are warranted for developing well-defined products with beneficial properties.

Although the Aloe leaf, commercially divided into the Aloe “inner leaf” gel and Aloe latex products [19,20], has been subjected to several scientific studies aimed to characterize the carbohydrate rich composition of Aloe gel [18,21] and the phenolic composition of the anthraquinone rich Aloe latex or yellow exudate [3], studies characterizing the metabolite profile of Aloe are limited. In this regard, metabolomics, the profiling of a wide range of small molecules in a sample, is starting to be utilized to identify and characterize the metabolite composition of plants and extracts [22,23]. While some recent studies have analyzed the metabolite profile of Aloe [24,25,26], these are limited and have not explicitly been associated with the gel component, nor to its effect on immune cells. Thus, knowledge regarding the metabolite composition of Aloe gel and any relationship with immune cell activity remains sparse.

We hypothesize that the effect of Aloe gel on blood T cell activity is due to a synergistic action of several compounds rather than linked to a specific component as suggested previously by some studies [4,9,10,12,18]. Therefore, this study determined the effect of various commercially available Aloe gel extracts on human blood T cell activity in vitro and correlated the effect to their standard phytochemical quality characteristics and metabolite composition, with the aim to provide an insight into the bioactive composition of Aloe.

## 2. Results

### 2.1. Standard Phytochemical Quality Characteristics of Aloe Extracts

In accordance with the Aloe quality standard requirements set by the International Aloe Science Council (IASC), all study extracts (Table 1) contained satisfactory amount of acemannan [27,28], i.e., at least 5% dry weight content and safe limit of anthraquinone [29,30] (≤10 ppm aloin) for Aloe products intended for oral consumption [28] (Table 2). While some marker components were not detected in extract CX03, the presence of high concentration of acemannan, unique to Aloe [31] and quantification of Ca and Mg [31], confirms its authenticity as an Aloe extract. Thus, the quantitative ^1^H-NMR and HPLC-UV analyses confirmed that all dehydrated Aloe extracts included in this study fulfilled requirements for satisfactory quality, with minimum degradation and no added preservatives or adulterants. The Aloe extracts, however, varied in their standard phytochemical quality characteristics, in particular the acemannan content, which varied from 9.8% to 91.4% in the different extracts (Table 2).

### 2.2. Aloe Extracts Vary in Their Effect on T Cell Activation and Proliferation, Which Is not Correlated to Their Acemannan Content

First, we compared the effect of Aloe extracts on T cell activation and proliferation. PBMCs were polyclonally stimulated for 3 days in the absence or presence of various Aloe extracts differing in their method of processing, product description and standard phytochemical quality characteristics (Table 1 and Table 2), at two different concentrations, 0.1 and 0.5 mg/mL (Figure 1). The frequencies of activated T cells and proliferating T cells were studied among viable 7AAD^−^ Annexin V^−^CD3^+^ T cells (gated as shown in Figure 1A).

When cultured in the presence of Aloe extracts at 0.1 mg/mL, the frequency of CD3^+^ T cells expressing the activation marker CD25, decreased in the presence of CX04 (Figure 1B). Along with the reduced T cell activity, flow cytometric analysis of CFSE-stained CD3^+^ T cells also showed that CX04 as well as CX01, reduced the proliferation of polyclonally stimulated T cells (Figure 1C). At the higher concentration (0.5 mg/mL), Aloe extracts CX01, CX02 and CX04 showed reduced T cell activity and proliferation of polyclonally stimulated T cells (Figure 1D,E). No such effect on T cell activation and proliferation was seen with extracts CX03 and CX05 at the concentrations tested (Figure 1B–E).

While the reduced T cell activation and proliferation could be attributed to extracts CX01, CX02 and CX04 with acemannan content ranging from 10–20%, CX05 with 10% acemannan and CX03 with high acemannan (91%) content showed no such effect. Furthermore, though we lack information regarding molecular weight for all extracts tested (Table 1), CX05 with a high molecular weight polysaccharide content (Table 1), showed no effect on T cell activation and proliferation (Figure 1). Thus, our results suggest that the effect of Aloe extracts on T cell activity and proliferation are not dependent on acemannan content, nor can it be attributed specifically to its high molecular weight polysaccharide content.

### 2.3. Aloe Extracts Induce a Concentration Dependent Effect on Apoptosis and Cell Death in Polyclonally Stimulated T Cells

Next, we determined the frequencies of apoptotic and dead cells, measured by Annexin V and 7AAD, respectively, among CD3^+^ T cells cultured in the presence of various Aloe extracts. None of the Aloe extracts had any effect on the frequency of Annexin V^+^ cells and 7AAD^+^ cells among the CD3^+^ T cells when tested at the lower concentration (0.1 mg/mL) (Figure 2A,B). At the higher concentration (0.5 mg/mL) an increase in Annexin V^+^ cells among the CD3^+^ T cell population, indicative of apoptotic cells, was seen in the presence of CX01 and CX05 (Figure 2C). Compared to the control (PHA alone), a slight increase in apoptotic cells from 0.3% (0.2–0.7%) to 3% (1.7–3.6%) in the presence of CX01, and 6% (3–8%) in the presence of CX05 were recorded. The frequency of 7AAD^+^ cells among CD3^+^ T cells, indicative of cell death, also increased in the presence of CX01, CX02 and CX05 at 0.5 mg/mL (Figure 2D). The extracts CX03 and CX04 neither influenced the frequency of Annexin V^+^ nor 7AAD^+^ cells amongst the CD3^+^ T cell population, at the two concentrations tested (Figure 2).

### 2.4. Aloe Extracts Differ in Their Metabolite Composition

We further investigated the metabolite composition of Aloe extracts, using Aloe extracts with minimum three available batches (Table 1). Overall, 210 individual metabolites were identified in Aloe extracts based on a combination of retention index and mass spectrum, covering compound classes including amino acids, organic acids, and mono- and disaccharides and their amine derivatives. A Principal component analysis (PCA) biplot of these metabolites showed a clear separation between the various Aloe extracts, indicating that each Aloe extract has its own distinct metabolite composition (Figure 3). While some minor batch differences were detected within each extract, the major driver of the separation was related to the unique metabolite composition of each extract (Figure 3).

Aloe extract CX01 had a metabolite profile rich in amines such as putrescine and aminobutyric acid; CX02 contained higher levels of diverse saccharides such as xylulose and rhamnose; CX04 had an enriched composition of organic acids such as carboxylic acid including malic acid; and CX05 was rich in phytosterol beta-sitosterol, when compared to the metabolite profile of the other analyzed Aloe extracts (Figure 3).

### 2.5. The Effect of Aloe Extracts on T Cell Activity and Proliferation Is Correlated to Their Distinct Metabolite Composition

Since variation in the effect of Aloe extracts on T cell activation and proliferation was not associated to their standard phytochemical quality characteristics, we explored whether the metabolite composition of the extracts might explain differences in T cell activation. A hierarchical cluster analysis based on the metabolite composition distinctly clustered Aloe extracts CX01, CX02 and CX04 from Aloe extract CX05 (Figure 4A). This clustering was in agreement with the immune activity of the extracts. Due to the suppressive effect on T cell activation and proliferation for extracts CX01, CX02 and CX04, this cluster was named “High Effect Cluster” while CX05, which showed no T cell effect, was termed “Low Effect Cluster” (Figure 4A). To determine the metabolites contributing most to the separation between the High and Low Effect Clusters, we carried out a discriminant orthogonal partial least squares-discriminant analysis (OPLA-DA) analysis which resulted in a highly robust and predictive model after feature reduction [R^2^Y = 0.97, Q^2^ = 0.95] including 25 metabolites (Figure 4B). While the High Effect Cluster was based on higher concentrations of several diverse sugars and sugar derivatives including 2-deoxy glucose, N-acetylglucosamine, inositol, ribose, sucrose, and arabinose, the Low Effect Cluster was mainly defined by beta-sitosterol (a plant sterol), the nucleoside adenosine and derivatives of the fatty acid octadecenoic acid (Figure 4C).

Additionally, we carried out PLS modelling and used the correlation matrix of the PLS plot to explore the relation between the metabolite composition of Aloe extracts and their effect of T cell activity. Many of the metabolites enriched in the High Effect Cluster including 2-deoxy-glucose, ribose-5-phosphate, n-acetyl-glucosamine and inositol were negatively correlated with T cell activity and/or T cell proliferation of Aloe (R < −0.4). On the contrary, metabolites defining the Low Effect Cluster such as adenosine and 9,12,15-Octadecatrienoic acid showed a positive correlation with T cell activity of Aloe (R > 0.4) (Table 3, Figure 4D).

## 3. Discussion

This study demonstrates that Aloe gel extracts with similar commercial description vary in their standard phytochemical quality characteristics as well as in their effect on human blood T cell activity in vitro. Further, each Aloe extract tested in this study had its own distinct metabolite composition, potentially causing the varying effect of Aloe extracts on T cell activity, which may at least partly explain the variability of Aloe extract bioactivity described in the literature. Overall, our results suggest that all commercially available Aloe extracts are unique and that substantive differences in their metabolite profiles lead to differential biological activity, which is independent of its acemannan content.

Aloe is one of the most popular medicinal plants used worldwide, nevertheless little work has been conducted on its composition or function through metabolomics. To the best of our knowledge, few previous studies have investigated the metabolite composition of Aloe gel extract [22,24,25,26], and none have compared commercially available Aloe extracts or correlated the metabolite profile to their effect on T cell activation and proliferation in vitro. This study is novel, for the first time demonstrating that a distinct metabolite profile of Aloe, rich in sugars and sugar derivates such as 2-deoxy-glucose, n-acetylglucosamine and glucosamine previously associated with modulatory effects on immune cells [32,33,34] correlated to effects on T cell activity and proliferation in vitro.

Metabolomic compositional differences seen among the various Aloe extracts analyzed in our study were notable. The majority of the metabolites identified for driving the differences between the various extracts were sugars, organic acids, phytosterols and amino acids, which are common constituents in all plants [35,36]. The most likely cause of these differences are the processing and extraction methods of Aloe extracts [18,27], though details on these processes are not readily available for commercial extracts. For example, among the metabolites that best defined extract CX02, the majority were identified as carbohydrates or carbohydrate derivatives that cover a wide range of molecules of similar polarity, which indicate a difference in the extraction selectivity, rather than a difference in the starting material. Further, the extracts with the greatest bioactivity had higher concentrations of glucuronic acid, rhamnose and galactose, previously reported to be present in Aloe gel [37]. Again, these results suggest that the metabolite composition is linked to extraction methods, which may selectively favor specific class of compounds.

T cell activation and proliferation is a complex process, involving a cascade of reactions and signals [38]. In this study the expression of CD25, the α chain of the IL-2 receptor, which binds to IL-2, a key cytokine involved in the activation, survival, expansion, and function of T lymphocytes, was used as the activation marker [39]. Although the addition of some Aloe extracts to polyclonally stimulated T cells resulted in reduction in activation and proliferation of T cells and extends previous in vitro studies of the potential immunosuppressive properties of Aloe [5,12,15,16,40,41], others showed no such effect. Hitherto, no previous study has demonstrated a correlation between the immunomodulatory effects of Aloe and its bioactive components or composition. Still, the immunosuppressive effects of Aloe has been attributed to its inner gel component [15,16], in particular the main polysaccharide acemannan [42,43], as well as Aloe sterols [41], and also high-molecular weight Aloe polysaccharide components [12]. In contrast, acemannan and other Aloe inner gel components have been associated with immunomodulatory, immunostimulatory or even undesirable antagonistic properties [3,44]. These divergent results emphasize the complex nature of the molecular constituents of Aloe.

Acemannan, which in our study ranged from 9.8 % to 91.4 %, often implicated to be related to the potency of Aloe, was not the determining factor for the varied response seen on stimulated T cells. Further, the effect of Aloe on T cells could not be attributed specifically to its high-molecular Aloe polysaccharide composition. However, those Aloe extracts that had a higher proportion of diverse mono- and disaccharides were more potent in reducing the T cell activation and proliferation. Hence, the distinct differences in the untargeted metabolite profile of Aloe extracts were linked to their effect on T cell activity. Further, the effect of Aloe on T cell activity was dependent on concentration, where certain extracts reduced T cell proliferation and abrogated T cell activation only when added at the high concentration (0.5 mg/mL). This is in line with previous results by us and others [5,13], suggesting that different concentrations of Aloe extracts may confer different effects and potency depending on different thresholds for bioactive components. Further, it can be argued that the reduced T cell activation and proliferation in the presence of Aloe extracts could be due to cell death and apoptosis. T cell activity and proliferation was determined among live and non-apoptotic CD3^+^ T cells and selective extracts influenced T cells also at the lower concentration but did not influence cell death or cell apoptosis. Nevertheless, the increased T cell death and apoptosis in the presence of the higher concentration of several Aloe extracts should be given attention. Altogether, our results suggest that concentration as well as the distinct metabolite composition of Aloe extracts need to be considered when developing well-defined products with bioactive properties.

While the difference in standard phytochemical characteristics of Aloe extracts failed to explain the variation in their effects on T cells, the hierarchical cluster dendrogram generated from Aloe metabolite profiles allowed for identifying two distinct cluster of extracts, which was in agreement with the immune activity of the extracts. The High Effect Cluster was enriched in 2-deoxy-glucose, n-acetylglucosamine and glucosamine, that previously have been associated with modulatory and suppressive effects on immune cells [32,33,34]. These metabolites, along with others making up the distinct composition of the extracts, may potentially cause the reduced T cell activity when added to the cell cultures. The negative correlation between the High Effect Cluster metabolites and T cell activity and proliferation, although should be interpreted with caution, further supports this theory. On the contrary, metabolites associated with the Low Effect Cluster showed moderately positive correlation with T cell activity. Although the Low Effect Cluster extract did not influence T cell activity in our study, the distinct metabolites of this cluster, adenosine and beta-sitosterol, have been shown to possess anti-inflammatory as well as immunostimulatory properties [45,46]. The absence of effect on T cell activity by the Low Effect Cluster could be due to sub-optimal concentration of these components in the extract. Our results support the importance of the synergistic effect of multiple components, rather than the effect of individual constituents, for bioactive potency. While not all the extracts compared in this study influenced T cell activity, several of the metabolites that differentiated the extracts have been implicated in human health [32,33,45,46]. Thus, even though each Aloe extract may possess beneficial properties, our results propose the importance of identifying and linking bioactive composition of Aloe extracts with the desired biological effects.

This study has both weaknesses and limitations. We have only looked at Aloe extracts through the lens of GC-MS, which is a relatively easy and quick method and provides a distinct set of metabolites as an appealing explanation for the variability of Aloe extracts on T cell activity. Still, there may be other vital compositional differences between the extracts such as peptides or larger polysaccharides that are not detected using the GC-MS method. Further, it should indeed be noted that due to the complex nature of monosaccharides, many with the same molecular weight, there is scope for misidentification. Glucosamine and acetylglucosamine, identified as important bioactive metabolites, not usually found in plant products, are likely products of fermentation. Thus, more extensive analyses using in depth metabolomic methods, including methods that can chromatographically resolve all individual monosaccharides as well as peptidomics are warranted to understand further the variation in Aloe extract composition due to different production processes. Further, while the reduction in activation and proliferation of polyclonally stimulated T cells in the presence of Aloe extracts may be indicative of its potential immunosuppressive property, the mode of action of Aloe remains unknown. It would be worthwhile to study further the effect of Aloe on immune activity to clarify mechanisms in more detail.

The present study provides an improved insight into the complex and synergistic bioactive composition of Aloe. The results also promote the importance of the methods for processing and extraction of commercially available Aloe extracts, as these procedures may imprint substantial differences in metabolite composition, making each extract unique and influence their effect on immune cell activity. Future focus should be directed towards exploiting metabolomics to improve the phytochemical characterization of Aloe extracts, enabling design and formulation of the next generation of functional foods and therapeutic products.

## 4. Materials and Methods

### 4.1. Plant Material—Aloe Extracts

Five dehydrated gel extracts of *Aloe barbadensis* Mill. obtained from Calmino group AB, Sweden, were included in the study (Table 1). The extracts, originating from different geographical locations, are inner leaf gel extracts with similar commercial description and the same ratio of solids to gel content (200:1 i.e., concentrated solids relative to fresh aloe inner leaf gel). All extracts are processed using different techniques and purified to remove majority of the anthraquinones (aloin and aloin-related compounds), components of the Aloe latex, which although implicated with the traditional intended use of Aloe for the short-term treatment of occasional constipation and well described in several monographs and pharmacopeia [47,48,49], have however recently been associated with safety concerns [11,50,51] Blended at various concentrations, these extracts make up commercially available Aloe inner gel extracts (Calmino group AB, Sweden), previously studied by our group [13].

### 4.2. Quality Assessment of Aloe Extracts and Quantification of Standard Aloe Marker Components

A commercially available certification method using quantitative proton nuclear magnetic resonance (^1^H-NMR) spectroscopy as previously described [31,52], was performed at Spectral Service AG (Köln, Germany), for authentication and quality assessment of Aloe extracts. Briefly, sample preparation was carried out by dissolving 50 mg Aloe extract mixed with 5 mg nicotinamide as an Internal Standard in 1000 μL Cs-EDTA stock solution. NMR measurements were performed on a Bruker Avance III 500 MHz spectrometer (Bruker BioSpin, Rheinstetten, Germany) and ^1^H-NMR spectra were recorded using a standard one-dimensional pulse sequence (zg30 program in Bruker language code). Finally, NMR spectra were automatically processed by phasing, baseline-correction, and integration using a C-program for TopSpin 3.2 (Bruker BioSpin) and MATLAB script (MATLAB 2015a; MathWorks, Natick, MA) [31]. Essential Aloe marker components, including acemannan, glucose, malic acid known to be naturally present in Aloe gel, and iso-citric acid (whole-leaf marker) were quantified by ^1^H-NMR [28,31]. Furthermore, ^1^H-NMR spectroscopy also assessed microbial, chemical and enzymatic degradation markers (lactic acid, acetic acid, succinic acid and fumaric acid), as well as presence of additives such as maltodextrin [27,31]. HPLC-UV, also performed at Spectral Service AG, was used for quantification of anthraquinones content (aloin and aloin-related compounds) in the Aloe extracts. UV spectra were recorded on a Shimadzu UV-1800 spectrophotometer (Shimadzu, Japan) in the spectral range of 190–400 nm with a resolution of 1 nm. The spectra of Aloe samples were decomposed by independent component analysis using the mutual information least dependent component analysis algorithm to enable quantitative determination of aloin in the Aloe samples [53].

### 4.3. Metabolite Analysis of Aloe Extracts

The metabolite profile of the Aloe extracts, including multiple batches of the same extract, was analyzed by using untargeted gas chromatography coupled mass spectrometry (GC-MS). Four out of five extracts, with multiple available batches were included in this analysis (Table 1). CX03 lacking multiple batches was not included in the metabolite analysis. Briefly, samples were extracted with water–methanol (1:9 *v*/*v*) containing ten stable isotope labeled internal standards followed by drying and derivatization by using oxymation and silylation. The derivatized extracts were injected into a GC-MS system (Shimadzu TQ8030) (Shimadzu Europa GmbH, Duisberg, Germany) and GC-MS scan data (50–750 *m*/*z*) was analyzed for targeted peak detection [54]. Peaks were identified based on a Matlab script and data was normalized based on the internal standard peak intensities [55].

### 4.4. Study Subjects and Sample Collection

Healthy individuals between the age of 18 and 60 years were recruited as study subjects, from whom venous blood was collected. The exclusion criteria were antibiotic treatment, usage of any Aloe products or treatment with any anti-inflammatory drug, such as ibuprofen or corticosteroids, during the last three months. Furthermore, none of the study subjects were suffering from any chronic disease with ongoing medication that could affect their immune system or induce risk of hemorrhage. The study was performed after receiving informed written consent from all study subjects, and the protocol was approved by the Regional Ethical Review Board in Gothenburg.

### 4.5. Analysis of the Effect of Aloe Extracts on Human Blood T Cell Activity In Vitro

Comparison of the effects of Aloe extracts, with regards to activation and proliferation of blood T cells from healthy human subjects, was carried out in vitro as described previously by our group [13]. Briefly, peripheral blood mononuclear cells (PBMCs) were isolated by density-gradient centrifugation on Ficoll-Paque (GE Healthcare Bio-Sciences AB, Uppsala, Sweden) and marked with 5,6-carboxyfluorescein diacetate succinimidyl ester (CFSE) (Invitrogen, Oregon, USA) at a final concentration of 0.2 mM, prior to being stimulated for 3 days. Polyclonal stimulation of PBMCs was carried out by adding phytohemagglutinin (PHA), lectin from phaseolus vulgaris, (Sigma-Aldrich, Germany) at a final concentration of 2 µg/mL, in the presence and absence of various Aloe extracts at different weight per volume concentrations (0.1 and 0.5 mg/mL).

Determination of T cell activity and proliferation was carried out by flow cytometry staining with a combination of the following antibodies: anti-CD3-PE-CF594 (clone UCHT1), anti-CD25-BV421, 7AAD and AnnexinV-APC (all from BD Pharmingen, San Jose, CA, USA) and analyzed using an LSR II flow cytometer (BD Pharmingen). T cells were identified based on their expression of CD3. T cell proliferation was measured by analysis of CFSE-emitting cells, apoptotic cells were defined as Annexin V^+^ cells among CD3^+^ T cells and dead cells were defined as 7AAD^+^ cells among CD3^+^ T cells. The frequency of activated T cells and proliferating T cells was defined as CD25^+^ cells and CFSE analyzed proliferating cells, among Annexin V^−^ 7AAD^−^ CD3^+^ T cells, respectively (gated as shown in Figure 1A). The data were analyzed using Flow Jo software (Treestar Inc., Ashland, OR, USA).

### 4.6. Statistical and Data Analyses

Differences between the effects of various Aloe extracts were evaluated by univariate statistical analysis, performed using GRAPHPAD Prism 9.0 (GraphPad Software, Version 9.3.1, La Jolla, CA, USA) and IBM SPSS Statistics for Windows, (Version 28.0.1.0, Armonk, NY, USA; IBM Corp). For comparisons between more than two groups, Kruskal–Wallis test followed by Dunn’s multiple comparison test was used. The Mann–Whitney U test was used when comparing two groups and the classical one-stage method was employed to correct for false-positive results in multiple comparisons, data are then presented as *q* values. *p* and *q* values <0.05 were considered as statistically significant. Data given in the figures and text are presented as median with interquartile range.

For the metabolomics data, principal component analysis (PCA) was conducted to determine overall variation among the Aloe extracts based on their metabolite profiles (X-variables), using the prcomp-function with z-score scaling, and visualized using the pca3d-package in R (version 4.2.0, Vienna, Austria) [56]. Hierarchical cluster analysis was performed on log transformed metabolomics data using the dendextend-package in R. Further, orthogonal partial least squares-discriminant analysis (OPLS-DA) was performed to determine metabolite profile differences between the Aloe extract clusters using SIMCA^®^ Software (version 16.0.1, MKS Umetrics AB, Umeå, Sweden). To refine the model, a variable influence on projection (VIP) was used to select variables based on discriminatory power within OPLS-DA modes. The quality of the OPLS-DA was determined based on the parameters R^2^, i.e., the goodness of fit of the model (values of ≥0.5 define good discrimination, where R^2^ = 1 is the best possible fit), and Q^2^, i.e., the goodness of prediction of the model (values of ≥0.4 are considered as good cross-validation) [57,58]. The reliability of the models was confirmed using analysis of variance testing of cross-validated predictive residuals (CV-ANOVA), where a *p* value < 0.05 indicated significantly different residuals of the compared groups. Additionally, OPLS-DA loading scatter plots were generated to identify the X variables (metabolites) most important for the discrimination between clusters and hence having large contributing to the model. Lastly, partial least squares projections to latent structures (PLS) was used to explore relationships between the metabolite composition of Aloe extracts and their effect of T cell activity. PLS loadings plot, refined using a variable influence on projection (VIP), was generated to indicate the importance or weight of the X-variables (metabolites), denoted by w, in explaining the modelling of Y-variables (T cell activity), denoted by c and plotted in a w*c plot. The results were further assessed via the correlation matrix of the PLS model using Pearson correlation [59]. *p* values were computed from correlation coefficients (R) using degree of freedom (df) based on a repeated measures design i.e., df = (k − 1) × (n − 1); where k denotes the number of factors analyzed and n denotes the number of subjects [60]. Thus, df = 6 × 3 = 18.

## Figures and Tables

**Figure 1 molecules-27-06643-f001:**
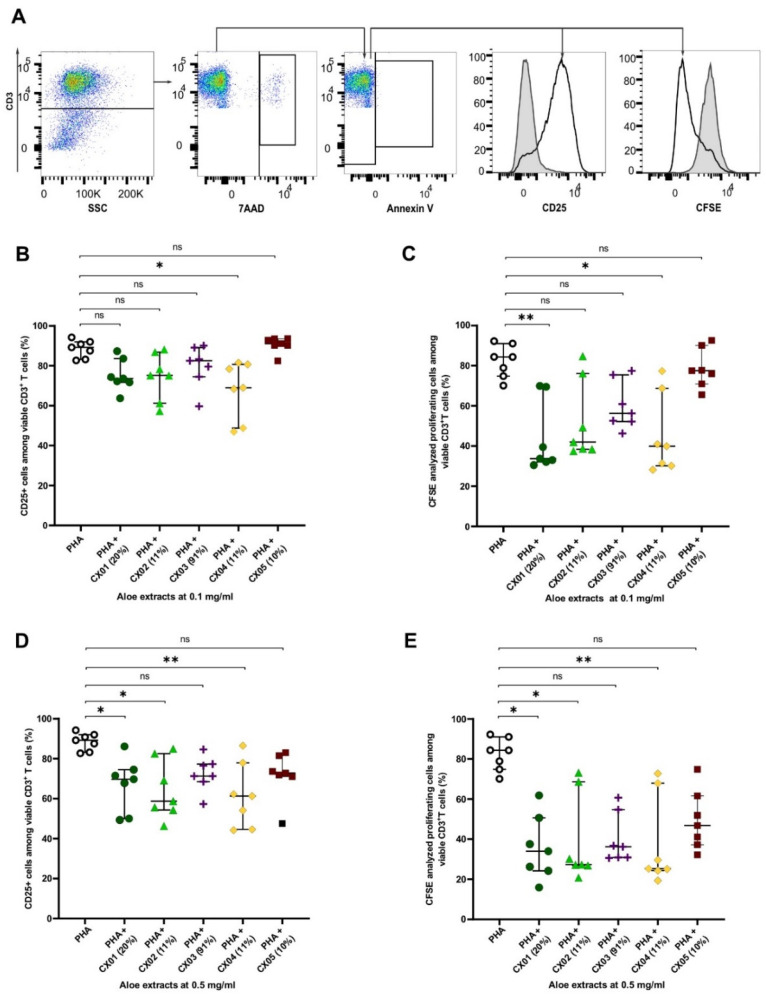
Effect of Aloe extracts on activation and proliferation of polyclonally stimulated T cells. Total PBMCs from seven donors were stimulated for 72 h with PHA in the presence or absence of Aloe extracts with varying acemannan content (% dry weight shown within brackets) at two doses (0.1 and 0.5 mg/mL). Extract CX05 is of high molecular weight. (**A**) Gating strategy for T cells identified as CD3^+^ cells. The frequency of activated T cells and proliferating T cells was defined as CD25^+^ cells and CFSE analyzed proliferating cells among viable 7AAD^−^ Annexin V^−^CD3^+^ T cells, respectively. Histograms show PHA stimulated (colorless) and unstimulated (gray solid, negative control) T cells with frequency of activated T cells (left) and proliferating T cells (right). (**B**,**D**) Activated T cells identified by the frequency of CD25^+^ cells among viable CD3^+^ T cells. (**C**,**E**) Proliferating cells among live CD3^+^ T cells measured by 5,6-carboxyfluorescein diactate ester (CFSE) emitting cells by flow cytometry. Asterisks represent significant *p* values: * <0.05; ** <0.01. ns: not significant.

**Figure 2 molecules-27-06643-f002:**
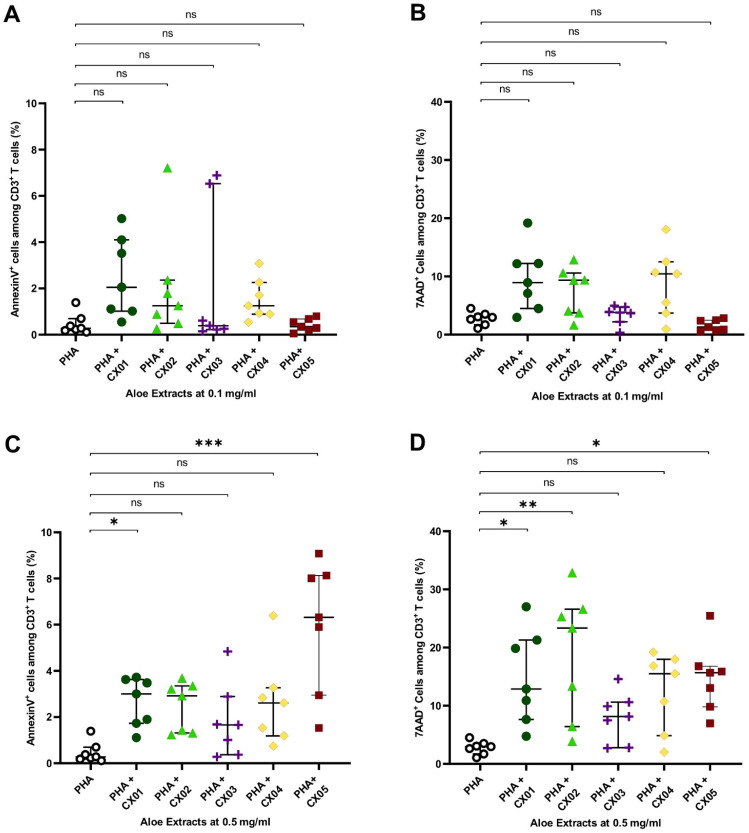
Effect of Aloe extracts on apoptosis and cell death in polyclonally stimulated T cells. Total PBMCs from seven donors were stimulated with PHA in the presence or absence of increasing concentrations of Aloe extracts (0.1 and 0.5 mg/mL). (**A**,**C**) The frequency of Annexin V^+^ cells among CD3^+^ T cells, indicative of apoptotic cells. (**B**,**D**) The frequency of 7AAD^+^ cells among CD3^+^ T cells, indicative of cell death. Asterisks represent significant *p* values: * <0.05; ** <0.01; *** <0.001. ns: not significant.

**Figure 3 molecules-27-06643-f003:**
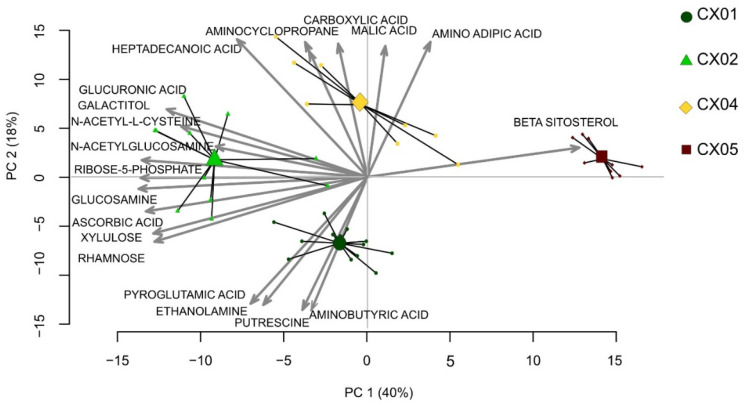
Metabolite profiles of various Aloe extracts. Principal component analysis (PCA) biplot derived from metabolites identified from various Aloe extracts, including CX01 (5 batches, dark green dots), CX02 (4 batches, light green triangles), CX04 (3 batches, yellow diamonds) and CX05 (3 batches, maroon squares). The individual samples of each extract based on 210 metabolites are connected to a centroid, indicating the weighted mean of the group. The PCA biplot also depicts the loading of the most relevant metabolites involved in the separation of the extracts.

**Figure 4 molecules-27-06643-f004:**
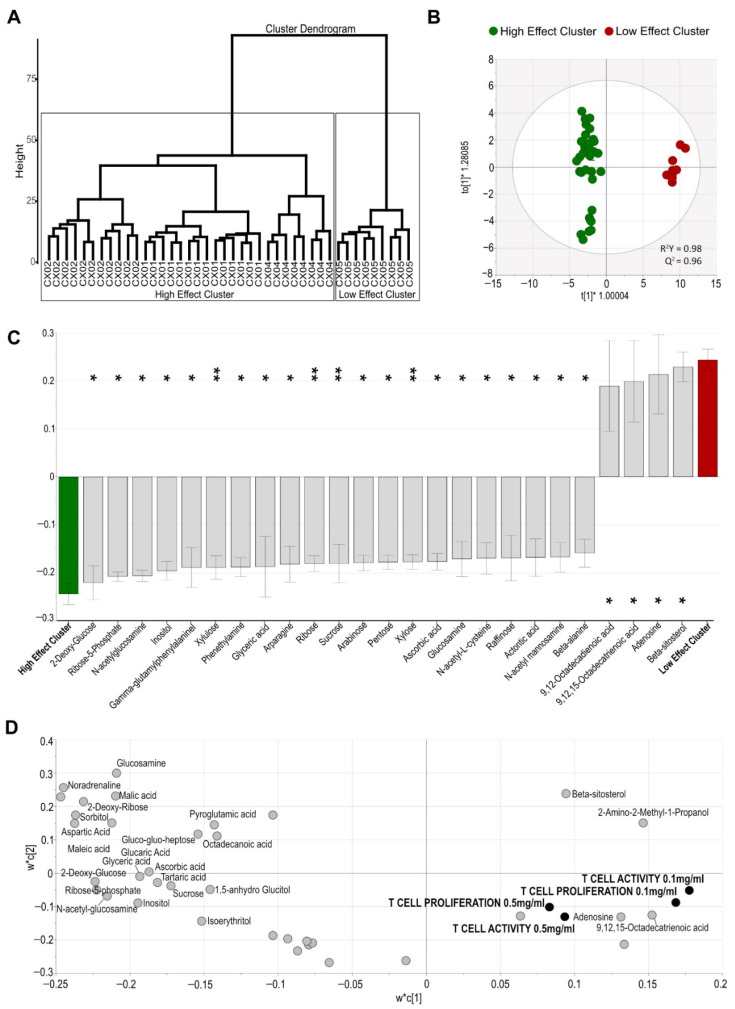
Aloe extract clusters based on their distinct metabolite composition associate to their immune cell activity. (**A**) Hierarchical cluster dendrogram for Aloe extracts based on their metabolite profile and identification of a High and a Low Effect Cluster. (**B**) Multivariate orthogonal partial least squares-discriminant analysis (OPLS-DA) score scatter plot showing discrimination between the High Effect Cluster compared to the Low Effect Cluster, based on the 25 most discriminatory metabolites (VIP > 1.2). (**C**) Metabolites differing between the Aloe extract clusters associated with High Effect and Low Effect. Error bars indicate 95% confidence intervals. (**D**) PLS loading scatter plot depicting the relation between the relative abundance of metabolites (light gray dots) and their effect on immune cell activity (black dots) in the presence of 0.1 and 0.5 mg/mL Aloe (VIP > 1.0). Asterisks represent significant *q* values: * <0.05; ** <0.01; VIP, variable influence on projection.

**Table 1 molecules-27-06643-t001:** Aloe gel extracts analyzed and compared in the study.

Extract	Batches	Processing	Product Description
CX01	2016014 ^a,b^ 2016011 ^b^ 2016007 ^b^ 2015016 ^b^ 2015007 ^b^	Freeze dried and decolorized	Not defined
CX02	20140402 ^a,b^ 20160603 ^b^ 20160606 ^b^ 20170902 ^b^	Freeze dried and decolorized	Not defined
CX03	20150326 ^a^	Freeze dried and decolorized	Not defined
CX04	20151203 ^a,b^ 20151107 ^b^ 20171003 ^b^	Spray dried and decolorized	Not defined
CX05	20160601 ^a,b^ 20171101 ^b^ 20170502 ^b^	Dehydrated, no decolorization	High molecular weight, Majority polysaccharides 50 kDa

^a^ Batch used for analyzing effect of T cell activity; ^b^ Batches used for metabolite analysis; NOTE: CX03 was not included in the metabolite analysis due to lack of minimum required batches.

**Table 2 molecules-27-06643-t002:** Standard phytochemical quality characteristics of Aloe gel extracts used in the study.

Component	Origin of Component	IASC Aloe Standard ^†^ Content by Dry Matter	Content % in Aloe Extracts				
			CX01	CX02	CX03	CX04	CX05
Acemannan	Fresh Aloe	≥5%	19.9	10.9	91.4	10.8	9.8
Glucose	Fresh Aloe	Present	25.1	27.7	Not detected	30.9	5.3
Fructose	Fresh Aloe	Present	3.8	7.3	Not detected	5.5	Detected
Malic acid	Fresh Aloe	Present	17.2	17.9	Not detected	19.8	10.9
Mg	Fresh Aloe	Present	Traces	Traces	0.1	0.8	0.1
Ca	Fresh Aloe	Present	4.4	4.1	0.8	3.0	2.5
WLM	Whole leaf marker	≤5%	2.5	Not detected	Not detected	Not detected	Not detected
Aloin	Anthraquinone	≤10ppm	<0.1 ppm	<0.1 ppm	<0.1 ppm	<0.1 ppm	<10 ppm
Citric acid	WLM or added acidifier	Limit <5% (undeclared)	2.7	1.3	Not detected	0.6	1.6
Lactic acid	Degradation (bacterial)	Limit <5%	1.9	2.7	Traces	2.2	0.6
Acetic acid	Degradation (chemical)	Limit <5%	Traces	Traces	Traces	Not detected	Not detected
Succinic acid	Degradation (enzymatic)	Limit <5%	Not detected	Not detected	Not detected	Not detected	Not detected
Fumaric acid	Degradation (enzymatic)	Limit <5%	Not detected	Not detected	Not detected	Not detected	Not detected
Maltodextrin	Adulterant/Additive	Absent (undeclared)	Not detected	Not detected	Not detected	Not detected	Not detected
Sodium benzoate	Added Preservative	Absent (undeclared)	Not detected	Not detected	Not detected	Not detected	Not detected
Potassium sorbate	Added Preservative	Absent (undeclared)	Not detected	Not detected	Not detected	Not detected	Not detected

NOTE: Content % data refers to dry matter; † Aloe Inner leaf products intended for oral consumption [27,28].

**Table 3 molecules-27-06643-t003:** The correlation between the relative abundance of metabolites and their effect on T cell activity and T cell proliferation.

Metabolites	T Cell Activity 0.1 mg/mL	T Cell Proliferation 0.1 mg/mL	T Cell Activity 0.5 mg/mL	T Cell Proliferation 0.5 mg/mL
Aspartic Acid	**−0.45**	**−0.62**	−0.28	−0.30
Noradrenaline	**−0.58**	**−0.62**	**−0.46**	−0.38
2-Deoxy-Ribose	**−0.52**	**−0.60**	−0.37	−0.35
Maleic acid	**−0.67**	**−0.56**	**−0.56**	−0.35
Sorbitol	**−0.51**	**−0.55**	−0.43	−0.36
Glucaric Acid	−0.37	**−0.54**	−0.37	−0.38
2-Deoxy-Glucose	**−0.53**	**−0.54**	−0.18	−0.21
Ribose-5-phosphate	**−0.50**	**−0.52**	−0.18	−0.21
Glucosamine	**−0.47**	**−0.50**	**−0.52**	−0.40
N-acetyl-glucosamine	**−0.53**	**−0.48**	−0.19	−0.18
Glyceric acid	**−0.49**	**−0.47**	−0.16	−0.17
Gluco-gluo-heptose	−0.19	**−0.45**	−0.10	−0.27
Malic acid	**−0.61**	−0.44	**−0.50**	−0.28
Ascorbic acid	−0.36	−0.43	−0.20	−0.23
Tartaric acid	**−0.52**	−0.42	−0.19	−0.13
Pyroglutamic acid	−0.15	−0.42	−0.14	−0.28
Inositol	**−0.54**	−0.41	−0.19	−0,13
Sucrose	**−0.55**	−0.35	−0.25	−0.11
Isoerythritol	−0.42	−0.34	−0.04	−0.05
1,5-anhydro Glucitol	**−0.49**	−0.31	−0.18	−0.07
Octadecanoic acid	**−0.51**	−0.30	0.44	0.20
Adenosine	**0.45**	0.23	0.44	0.18
2-Amino-2-Methyl-1-Propanol	0.41	0.28	0.10	0.05
9,12,15-Octadecatrienoic acid	**0.48**	0.28	−0.36	−0.15

Light gray and dark gray highlighted cells represent negative (−0.4 < *r* < 0) and positive (0 < *r* < 0.4) correlations, respectively. Significant correlations are displayed in bold. Note: Metabolites that have a high contribution to the T cell activity and T cell proliferation were determined by correlation matrix of the PLS plot (VIP > 1.0).

## Data Availability

The data presented in this study are available from the corresponding author on reasonable request.

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
