# Peer review of "Differences in Metabolite Composition of *Aloe barbadensis* Mill. Extracts Lead to Differential Effects on Human Blood T Cell Activity In Vitro"

_molecules, 2022, doi:10.3390/molecules27196643_

Round 1

Reviewer 1 Report

Improve English and and  some colourful pictures 

Author Response

Response to Reviewer 1 Comments

Point 1: Improve English and and some colourful pictures

Response 1: Thank you for the comment and constructive suggestion. We have now updated and improved English throughout the manuscript, as well as added colour to all the Figures.  

Reviewer 2 Report

Manuscript ID: molecules-1930473

Title: Differences in metabolite composition of Aloe barbadensis Mill.

extracts lead to differential effects on human blood T cell activity in vitro

The metabolite composition of various commercial Aloe extracts and assess their effects on human blood T cell activity in vitro was presented.

However, some details should be improved and explained. I would like to make some comments that authors could take into account to improve the overall quality of the manuscript.

Comments:

Tab. 1 – The title not describes properly content of this table.

Fig. 3 – it is enough to show only biplot (Fig. 3B) which contain all data presented previously in Fig. 3A.

Fig. 4 – The extract CX08 was included into Fig. 4A what is not consistent with Tab. 1.

Fig. 4B – It is hard to distinguish high and low effect clusters.

Fig. 4 – The statistical analysis was performed including “T CELL ACTIVITY 0.1 mg/ml” and “T CELL PROLIFERATION 0.1 mg/ml” (PLS, Fig. 4D). I do not understand why T cell activity and proliferation at 0.5 mg/ml was not included into PLS, OPLS-DA, cluster analysis. Are these results (presented in Fig. 4) consistent with Fig. 1?

Tab. 3. It is not known which correlation are statistically significant. Are these correlations calculated for data recorded at 0.1 and/or 0.5 mg/ml?

Lines 384-385: Lack of details about HPLC-UV, any methodology or literature cited.

General comment: The targeted GC-MS analysis was performed but I cannot see justification for that, especially if metabolite profile was correlated with T cell activity and proliferation. The reasonable approach is to perform untargeted analysis.

Author Response

Point 1: Tab. 1 – The title not describes properly content of this table.

Response 1: We thank the reviewer for this attention to detail and have corrected the title for Table 1 to describe the content more appropriately (page 3).  

Point 2: Fig. 3 – it is enough to show only biplot (Fig. 3B) which contain all data presented previously in Fig. 3A.

Response 2: We agree with the reviewer and as suggested, have now updated Figure 3 and the Figure 3 legend, to depict the most valid information using only the PCA biplot (page 7). The corresponding result section 2.4 on pages 4 and 7 has also been updated.

Point 3: Fig. 4 – The extract CX08 was included into Fig. 4A what is not consistent with Tab. 1.

Response 3: We again thank the reviewer for this attention to detail and have corrected Figure 4A (page 8) to depict the extracts consistently used throughout study, including CX01-CX05.

Point 4: Fig. 4B – It is hard to distinguish high and low effect clusters.

Response 4: Thank you for this observation. We have updated Figure 4B (page 8), which is now shown in colour to enable clear distinction between the high and low effect clusters.

Point 5: Fig. 4 – The statistical analysis was performed including “T CELL ACTIVITY 0.1 mg/ml” and “T CELL PROLIFERATION 0.1 mg/ml” (PLS, Fig. 4D). I do not understand why T cell activity and proliferation at 0.5 mg/ml was not included into PLS, OPLS-DA, cluster analysis. Are these results (presented in Fig. 4) consistent with Fig. 1?

Response 5: Thank you for this valid question. The results presented in Figure 4A to 4C, including the cluster analysis and OPLS-DA were based only on the metabolite composition of Aloe extracts. The clustering seen in Figure 4A was however in agreement with the immune cell activity of the Aloe extracts, seen in Figure 1. Which then lead us to carry out a PLS depicting relation between the metabolite composition of Aloe extracts and their effect of T cell activity. To further clarify this, we have now updated the result section 2.5, page 7 (line 240-241).

We however completely agree with the reviewer and to improve the consistency in this manuscript, we have now included the results for the T cell activity and proliferation at 0.5 mg/ml in Figure 4D (page 8) as well as in Table 3 (page 9), described in the next point.

Point 6: Tab. 3. It is not known which correlation are statistically significant. Are these correlations calculated for data recorded at 0.1 and/or 0.5 mg/ml?

Response 6: Thank you for this comment. As pointed out by the reviewer, our manuscript had missing information regarding the correlations calculated and their significance. We have now added results depicting correlations between the metabolites and the T cell activity and T cell proliferation at both the Aloe concentrations used (0.1mg/ml and 0.5mg/ml). Several of these correlations were also statistically significant (P < 0.05). This information, displayed in bold, has now been added in Table 3 on page 9. P values were computed from correlation coefficients (R) using degree of freedom (df) based on a repeated measures design i.e., df = (k−1) x (n−1); where k denotes the number of factors analysed and n denotes the number of subjects. Thus, df = 6 x 3 = 18. This information has been added to the Materials and Methods section 4.6, page 14 (lines 551-554).

Point 7: Lines 384-385: Lack of details about HPLC-UV, any methodology or literature cited.

Response 7: We understand and appreciate the reviewer’s comment. We have now added more detailed information regarding the HPLC-UV in the Materials and Methods section 4.2, page 12 (lines 461 – 465), along with relevant cited literature.

Point 8: General comment: The targeted GC-MS analysis was performed but I cannot see justification for that, especially if metabolite profile was correlated with T cell activity and proliferation. The reasonable approach is to perform untargeted analysis.

Response 7: Thank you for this comment. The metabolomics analyses were indeed conducted using GC-MS in scanning mode – i.e., untargeted data acquisition. We briefly mention the untargeted nature of this analysis in our abstract (lines 23-24), however, to clarify this we have now updated the Materials and Methods section 4.3, page 12 (line 469).

The data were processed using a script which uses retention time, overall spectrum, and the fragment ion to perform compound identification. We have used this approach rather than general peak deconvolution including unknowns/unidentified features as the unidentified features do not give much information about biological mechanisms. We have done a comparison between targeted feature extraction and untargeted feature extraction followed by library matching, and there is a large overlap with the former being more effective. Hence, we have used this ‘targeted untargeted metabolomics’ strategy for the evaluation of metabolite profile in this study.

Reviewer 3 Report

The article by Ahluwalia et al. entitled “Differences in metabolite composition of Aloe barbadensis Mill. extracts lead to differential effects on human blood T cell activity in vitro” was submitted to Biomarkers. The reviewer's enthusiasm remains limited due to the following concerns

1.     The authors should provide their justification and relevance of the study. Authors have earlier published similar studies in the journal (J Ethnopharmacol. 2016 Feb 17;179:301-9. doi: 10.1016/j.jep.2016.01.003. Epub 2016 Jan 4. PMID: 26771068.), thus, there is no novelty of the present study.

2.     Most of the cited references in this study were outdated. Authors can replace the references with the last 5 years

3.     Minor typographical errors were found throughout the manuscript and should be amended.

4.     The authors did not perform any animal model experiments in the present study.

5.     Authors can validate their study using apoptosis markers and cell proliferation markers

Author Response

Point 1: The authors should provide their justification and relevance of the study. Authors have earlier published similar studies in the journal (J Ethnopharmacol. 2016 Feb 17;179:301-9. doi: 10.1016/j.jep.2016.01.003. Epub 2016 Jan 4. PMID: 26771068.), thus, there is no novelty of the present study.

Response 1: We thank the reviewer for this reflection. As compared to the previous study mentioned by the reviewer, we think that our current study using metabolomics provides a novel and improved insight into the complex and synergistic bioactive composition of Aloe. We agree with the reviewer, that both the previous published study mentioned, and our current study did indeed determine the effect of various commercially available Aloe gel extracts on human blood T cell activity in vitro. However, as described in the Introduction section, page 2 (lines 70-80), no previous studies have analysed the metabolite profile of Aloe gel, neither have studies explicitly been able to associate composition/ components of Aloe gel to its effect on immune cells. Further, several previous studies have attributed the beneficial properties of Aloe to its polysaccharides (acemannan). Thus, we believe that our study demonstrating for the first time that a distinct metabolite profile of Aloe gel, independent of its acemannan content, could be correlated to its effects on T cell activity and proliferation in vitro, and thus adds novelty to this study. We discuss the novelty of this study in the Discussion section, page 8 (lines 316 – 324). Despite of this, we agree that the rationale for this study may have not been clearly described and we have now updated the aim of our study in the Introduction section, page 2 (lines 86-87), to further clarify this.

Point 2: Most of the cited references in this study were outdated. Authors can replace the references with the last 5 years

Response 2: We understand and appreciate the reviewer’s constructive comment and have now made relevant updates in our References. However, since the field of Aloe is not entirely new, we would like to keep some older references which have been pioneering in the field with well-established results.

Point 3: Minor typographical errors were found throughout the manuscript and should be amended.

Response 3: We thank the reviewer for this attention to detail and have made required amendments throughout the manuscript.

Point 4: The authors did not perform any animal model experiments in the present study.

Response 4: We appreciate the reviewer’s input on this matter. While animal studies would indeed be able to add exceptional value to this research topic, they were however beyond the scope of this manuscript. Future studies using animal models would greatly enhance our existing knowledge regarding the effect of Aloe gel on immune activity.

Point 5: Authors can validate their study using apoptosis markers and cell proliferation markers

Response 5: Thank you for this valid input. In agreement with the reviewer’s comment, our study has used well established apoptosis marker, AnnexinV-APC, as well as cell proliferation marker - 5, 6-carboxyfluorescein diacetate succinimidyl ester (CFSE), described in the Materials and Methods section 4.5, page 13. This is also depicted in our results sections 2.2 and 2.3, pages 3-4.    

Round 2

Reviewer 2 Report

The paper has been corrected significantly and I think that final version of this paper can be considered for publication.

Reviewer 3 Report

The present form is accepted for publication.